# Comparison of the Segmentation Results of Two Carrier Tracking Loop Types and Analysis of Theoretical Influencing Factors

**Qian Wang, Mengyue Han \*, Yuanlan Wen, Min He and Xiufeng He** 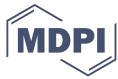

School of Earth Sciences and Engineering, Hohai University, Nanjing 210098, China; wqaloha@hhu.edu.cn (Q.W.); 20180812@hhu.edu.cn (Y.W.); mhe@hhu.edu.cn (M.H.); xfhe@hhu.edu.cn (X.H.)

\* Correspondence: myhan@hhu.edu.cn; Tel.: +86-15951936312

**Abstract:** This paper proposes an accurate quantitative segmentation method by analyzing the probability distribution of tracking variance and strict derivation based on the tracking loop theory. The segmentation points are taken as characteristics of phase lock loop (PLL) and frequency lock loop (FLL) performances, and the two factors that cause the performance difference are discriminator gain and filtering coefficient, which denote proportional and integration coefficients, respectively. The filtering coefficients lead to a difference of 2.5 dB-Hz between the FLL and PLL. Moreover, through the analysis of the normalized bandwidth and phase margin, it is found that the integration time and bandwidth need a dynamic balance to achieve the best performance. Finally, the simulation results and real data are in good agreement with the theoretical analysis results. The minimum mean error rate of the deviation between the real data and the theoretical data is only 1.8%. In the proposed method, the influence of external hardware factors on the tracking loop is removed, and the loop design factors are modeled directly. Instead of testing the denoising performance based on the ranging and angle measuring error after location calculation, the filter coefficient is proposed to evaluate the processing performance of the tracking loop objectively and directly at the theoretical level, which proposes a new performance evaluation method at the theoretical level. The results presented in this study provide theoretical support for the design of a new-type tracking loop with enhanced performances.

**Keywords:** GNSS; filtering coefficient; frequency lock loop; phase lock loop; forward loop segmentation

## 1. Introduction

Frequency lock loop (FLL) and phase lock loop (PLL) are two forms of carrier tracking loop. The FLL and PLL have similar structures that include discriminator, filter, and numerically controlled oscillator (NCO). However, their feedback parameters are different; namely, the FLL adjusts the frequency difference, and the PLL adjusts the phase difference. Hence, FLL and PLL have different performances. There have been many studies on FLL and PLL, most of which aim at improving the structure of one of the loops or combining the two loops to improve the tracking performance.

Some studies have analyzed the basic structure of the FLL or PLL and proposed some improved tracking loop design methods to enhance the tracking performance. Curran et al. [1] used the FLL to analyze in detail the characteristics of four commonly used frequency discriminators and their effects on the overall loop performance. Han et al. [2] studied the all-phase mathematical model of the typical digital frequency locked loop (DFLL) in the Z-domain and compared the dynamic performance of the DFLL with that of the analog frequency locked loop (AFLL). Mo et al. [3] proposed an algorithm for the FLL assisting the PLL-based fuzzy control, which can automatically switch between a pure Kalman filter (KF)-based PLL, pure KF-based FLL, and KF-based FLL-assisted PLL and can automatically adjust the noise bandwidth. Chen at al. [4] proposed and implemented an

adaptive joint vector phase lock loop (VPLL), which can improve the phase tracking performance under highly dynamic conditions. The test results have shown that compared with conventional VPLL, the proposed adaptive joint VPLL can improve the carrier phase tracking performance under highly dynamic conditions. Guo et al. [5] investigated the effects of the global navigation satellite system (GNSS) receiver tracking loop tuning on scintillation monitoring and estimated the PLL tracking jitter using simulated GNSS data. The results showed that receiver tuning had a minor effect on the scintillation indices calculation.

In order to improve the tracking loop performance, many studies focus on the structure modification, among which using KF is an effective method. Jiang et al. [6] analyzed the factors affecting the KF-PLL performance and proved the equivalence between the third-order PLL and the steady-state KF, thus demonstrating that the PLL is comparable to the KF. Yang et al. [7] presented a generalized multi-frequency carrier tracking structure, which integrated a conventional single-frequency independent tracking mode, a multi-frequency joint tracking mode, and a multi-frequency optimal tracking mode via an aggregate KF. Dou et al. [8] implemented three FLL types, i.e., the scalar FLL, the weighted least square-based vector FLL, and the extended KF-based vector FLL (EKF-VFLL). The results showed that in a highly dynamic environment, the advantages of the EKF-VFLL were more prominent than those of the other two methods. Cheng et al. [9] proposed an adaptive strong tracking Kalman filter (STKF) to further enhance the carrier tracking performance. The proposed algorithm was implemented in the software receiver and test results demonstrated that the proposed method had superior tracking performance over the general carrier tracking method in challenging environments.

There have been numerous studies on error segmentation in order to improve the positioning accuracy of GNSS receivers, but most of these methods are for data post-processing, and are rarely combined with the tracking loop structure. Zhang et al. [10] proposed a real-time adaptive weighting model to mitigate the site-specific unmodeled errors of code observations. The authors mainly used the difference between the carrier-to-noise ratio (CNR) estimated by the template function and its nominal value to divide the errors into two categories. When the difference between the CNR and the nominal CNR was smaller than the threshold, the site-specific unmodeled errors were considered negligible; otherwise, the unmodeled errors were considered significant. Fu et al. [11] developed a robust, combined, multiple-system precise point positioning (PPP) method, where the outliers were removed according to the unit weight standard deviation (STD) and the maximum residual of observation data. This method calculates the unit weight STD and maximum residual after obtaining the initial estimation result of the PPP, and then judges whether the removal conditions are met, which is mainly carried out during data processing. Lyu et al. [12] proposed a new multi-feature support vector machine (SVM) signal classifier-based weight scheme for GNSS measurements. The covariance of the non-line-of-sight (NLOS) pseudorange and phase observation error was categorized into three groups based on the NLOS error threshold. Cortes et al. [13] evaluated the performance and complexity of the state-of-the-art adaptive scalar tracking techniques used in modern digital GNSS receivers. The tracking variance of the loop was divided into four groups based on the difference between the estimated and actual bandwidths. The results showed that techniques achieved superior static and dynamic system performance were 1.5 times more complex than the traditional tracking loop.

The above-mentioned segmentation-related research mostly relies on a priori template function, which is easily constrained by environmental changes or is segmented by analyzing observation data residuals, which is generally completed at the data level, resulting in poor accuracy and delayed response. In addition, most of the aforementioned studies either consider the structure modification of only one loop type (FLL or PLL) to improve the performance or combine FLL and PLL using the fundamental theory. Further, most of them test the denoising performance of a tracking loop based on errors of ranging and angle measurement and integrate the influence of external hardware factors on the loop, so there are no objective evaluation indexes of a tracking loop's accuracy and robustness.

Therefore, to further analyze the causes of differences between FLL and PLL differences in theory, as well as to design a more robust tracking loop, this paper proposes an accurate quantitative segmentation method based on analyzing the probability distribution of tracking variance at the signal level. The tracking loop design factors are modeled directly, and filtering coefficients are introduced to evaluate the processing performance of a tracking loop objectively.

The main contributions of this work are as follows:

(1)    An accurate quantitative loop segmentation method is proposed, which provides a theoretical basis for the realization of robust fusion directly at the signal level in a new-generation tracking loop design. The segmentation results show the performance differences between the FLL and PLL;

(2)    The influence of external hardware factors on the loop is removed, and the loop design factors are modeled directly. The analysis results show that the integration time and bandwidth affect the segmentation results of both FLL and PLL, while the gain and filter coefficients result in a difference between the PLL and FLL.

The rest of the paper is organized as follows: In Section 2, the segmentation results of the PLL and FLL are introduced. In Section 3, the theoretical results are verified by the simulation, and the factors affecting segmentation are analyzed. In Section 4, the filter coefficients are discussed, and real data are used to validate the equations of filter coefficients. Finally, the conclusions are given in Section 5.

## 2. Materials and Methods

### 2.1. Derivation of Segmentation Points

The purpose of loop segmentation is to determine the strong and weak signal ranges of the FLL and PLL. The segmentation is based on the fact that the loop variance has different values at different values of the signal-to-noise ratio (SNR). Curran et al. [1] analyzed the FLL in detail, and their analysis results are used in this study to calculate the segmentation points of the FLL. Following the analysis principles [1], the loop variance and filter design of the PLL are derived and analyzed in detail, and the segmentation points of the PLL are calculated.

### 2.2. FLL Analysis Based on Related Work

This study takes the four-quadrant arctangent frequency discriminator as an example due to its large linear working range [1,14], so the variance of discriminator error can be expressed as [1]:

$$R_0^n = \frac{1}{T_L^2} \int_{-\pi}^{\pi} \int_{-\pi}^{\pi} f(\phi_2 - \phi_1)^2 P(\phi_1) P(\phi_2) d\phi_1 d\phi_2 \tag{1}$$

where $f(x)$ represents the functional relationship between input and output signals of a noise-free carrier frequency discriminator, thus, $f(x)$ represents the mapping of the true phase advance per sample period to the discriminator's noise-free estimation. $\phi_2$ denotes phase difference caused by the thermal noise at time $m$, and $\phi_1$ denotes a phase difference caused by the thermal noise at time $(m - 1)$. $T_L$ denotes the correlation period; $P(\phi)$ represents the probability density function (PDF) of the phase difference caused by the thermal noise, and it is given by:

$$P(\phi) = \frac{e^{-\frac{SNR}{2}}}{2\pi}\left(1 + \sqrt{\frac{\pi SNR}{2}}\cos(\phi)e^{\frac{SNR\cos\phi^2}{2}}\bullet x\right), x = \left(1 + erf\left\{\frac{SNR\cos\phi}{\sqrt{2}}\right\}\right), -\pi < \phi < \pi \tag{2}$$

The thermal noise is the main error source of a discriminator, and in general, the oscillator noise can be ignored. The PDF given by Equation (2) is closely related to the SNR [15] and their relationship is shown in Figure 1, where $\varphi$ denotes the phase difference.

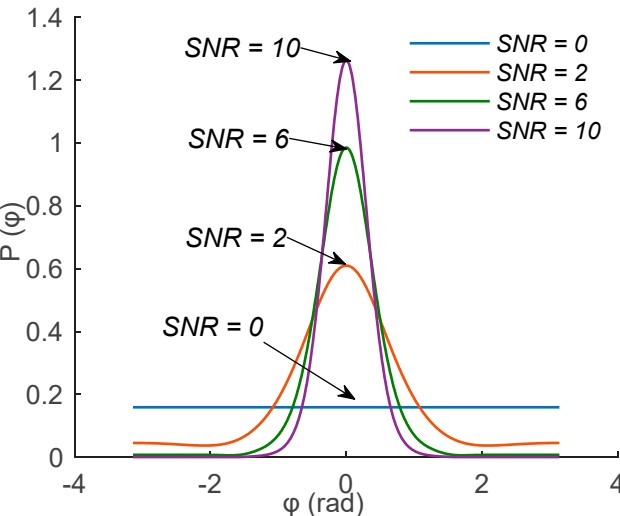

**Figure 1.** Relationship between the PDF and the phase difference under different SNR values.

As shown in Figure 1, with the decrease in the SNR value, the Gaussian trend of the PDF becomes less obvious, and finally, at the SNR of zero, the probability distribution obeys uniform distribution. Thus, the characteristics of the PDF can be used to determine the ranges of strong and weak signals of a tracking loop.

At a high SNR value, the variance can be expressed as [1]:

$$R_0^n = \frac{2}{SNR * T_L^2} \left( (\text{rad}/s)^2 \right) \tag{3}$$

At an extremely low SNR value, the PDF follows the uniform distribution in the range $[-\pi, \pi]$ [16], and its variance tends to $\frac{\pi^2}{3T_L^2}$ [16], which is a theoretical upper bound of the value calculated by Equation (1). In theory, when SNR is zero, the variance reaches the theoretical upper bound, which is not in line with actual situation. In fact, when the SNR is different from zero, the tracking variance is large, and the tracking loop has lost lock, so using this theoretical upper bound is not practical.

In order to simulate an actual situation, in this work, the tracking threshold value is taken as an upper bound value of Equation (4). In practice, the FLL tracking loops have a certain tracking threshold [17]. A conservative estimation method of the FLL tracking threshold is that three times of the mean square error of frequency measurement should not exceed a quarter of the range of frequency discrimination, which can be expressed as $3\sigma_{FLL} \le \frac{1}{4T_L}$ [17], where $\sigma_{FLL}^2$ represents the tracking variance of frequency discriminator, which is $R_0^n$ in this paper. The difference between $\sigma_{FLL}^2$ and $R_0^n$ is that the unit of $R_0^n$ is $(\text{rad}/s)^2$, while the unit of $\sigma_{FLL}^2$ is $Hz^2$.

When the threshold is reached, the SNR is considered being low, after unifying the unit, which is expressed as [18]:

$$\frac{9R_0^n}{4\pi^2} \le \frac{1}{16T^2} \tag{4}$$

Next, the variance of the second-order loop after filtering can be expressed as [1]:

$$\sigma_{\delta\omega}^2 = \frac{(R_0^n + 2R_1^n)T_L(2A_0^2 K_D + A_1(2 + A_0 K_D T_L))}{A_0 K_D^2(4 - K_D T_L(2A_0 + A_1 T_L))} - \frac{2R_1^n K_D T_L^2(A_1^2 T_L + A_0 K_D(A_0 + A_1 T_L)(2A_0 + A_1 T_L))}{A_0 K_D^2(4 - K_D T_L(2A_0 + A_1 T_L))} \tag{5}$$

where the relationship between $R_0^n$ and $R_1^n$ is given by [1]:

$$R_1^n \approx -\frac{1}{2}(1 - e^{-0.4864SNR})R_0^n \tag{6}$$

In Equation (5), $K_D$ denotes the frequency discrimination gain, which is calculated by [1]:

$$K_D^{A \tan 2} = 1 - 4\pi \int_{-\pi}^{\pi} P(\theta) P(\theta - \pi) d\theta \tag{7}$$

where $\theta$ represents the phase difference between the received signal and the local signal. In Equation (5), the $A_0$ and $A_1$ represent the proportion and integration gains, respectively. The values of $A_0$ and $A_1$ with respect to $\beta$ and $\eta$ can be obtained as follows [1]:

$$A_0 = \frac{1 - e^{-2\beta}}{K_D T_L} \tag{8}$$

$$A_1 = \frac{e^{-2\beta} \left(1 - e^{\beta(1+\eta)}\right) \left(1 - e^{\beta(1-\eta)}\right)}{K_D T_L^2} \tag{9}$$

where $\beta$ represents the oscillation fading factor, and $\eta$ is the oscillation damping factor. Generally, a quadratic polynomial $\beta \approx \frac{4}{\pi} B_\omega T_L - \frac{1}{6}(B_\omega T_L)^2$ is used for approximation of the analytical solution to $\beta$ [1], where $T_L$ denotes the integration time, and $B_\omega$ denotes the noise bandwidth. The damping coefficient is expressed as $\eta^2 = -1$.

By using the relationship between SNR and CNR, in all the above formulas, SNR can be transformed into CNR. The theoretical value of variance is given by Equation (1), and at a high CNR value, variance is expressed by (3). When Equation (1) and Equation (3) are substituted into Equation (5) respectively, an equation with unknown CNR is obtained. Thus, the value of the first segmentation point can be calculated. The theoretical variance deviates from the variance at high CNR. When the integration time bandwidth are 2 ms and 25 Hz, respectively, the CNR of the segmentation point is 48.496 dB-Hz [19].

By substituting different values of $R_0^n$ in Equations (1) and (4) into Equation (5), the second segmentation points can be calculated by solving this equation. The second segmentation point indicates that the theoretical variance reaches the upper bound when the CNR is 25.079 dB-Hz [19].

The segmentation result can be expressed as follows:

$$\sigma_{\delta\omega}^2 = \begin{cases} \frac{2c_1 + (2c_1 - c_2)\left(e^{-0.4864 SNR} - 1\right)}{c_3 K_D^2 T_L^2 SNR}, \\ \text{CNR} > 48.496 \text{ dB-Hz} \\ \frac{c_1 \frac{1}{(12T)^2} + \frac{1}{2 \cdot (12T)^2}(2c_1 - c_2)\left(e^{-0.4864 SNR} - 1\right)}{c_3 K_D^2 T_L^2}, \\ \text{CNR} < 25.079 \text{ dB-Hz} \end{cases} \tag{10}$$

where:

$$c_1 = T_L(2A_0^2 K_D^2 + 2A_1 K_D + A_0 A_1 K_D^2 T_L) \tag{11}$$

$$c_2 = 2T_L^2 \left(A_1^2 K_D^2 T_L + \left(A_0^2 K_D^2 + A_0 A_1 K_D^2 T_L\right)(2A_0 K_D + A_1 K_D T_L)\right) \tag{12}$$

$$c_3 = A_0 K_D(4 - T_L(2A_0 K_D + A_1 K_D T_L)) \tag{13}$$

*2.3. PLL Analysis*

The PLL is another type of carrier tracking loop whose purpose is to lock the phase of the input signal. According to the method of calculating the segmentation results of the FLL, the variance of a phase discriminator, filter coefficient, and segmentation results of the PLL can be obtained, and they are introduced hereinafter.

2.3.1. Phase Discriminator Variance

In order to compare the performance of the two loops, the four-quadrant arctangent discriminator (ATAN2) is used. The phase difference between the duplicated carrier signal and the received signal $e_m$ of the phase detector output [20,21] is expressed as:

$$e_m = K_D \delta\theta_m + n_m^\theta \tag{14}$$

where $K_D$ denotes the gain of the phase discriminator, $\delta\theta_m$ is the phase difference between times $(m-1)$ and $m$, and $n_m^\theta$ denotes the noise. The mean value of Equation (14) is given by [21]:

$$\mu_e = \int_{-\pi}^{\pi} f(\delta\theta + \phi)P(\phi)d\phi \tag{15}$$

The output of the ATAN2 is given by [21]:

$$f_{ATAN2} = \arctan2(I_m, Q_m) \tag{16}$$

where $I_m$ and $Q_m$ represent the in-phase and quadrature components of the input signal after correlation demodulation.

By substituting Equation (16) into Equation (15), the mean value of the output of ATAN2 can be obtained as follows [21]:

$$\begin{aligned} \mu_e^{A\tan 2} &= \int_{-\pi}^{\pi} \arctan(\cos(\delta\theta + \phi), \sin(\delta\theta + \phi))p(\phi)d\phi \\ &= \int_{-\pi}^{\pi} \phi p(\phi - \delta\theta)d\phi \end{aligned} \tag{17}$$

Based on Equation (17), and using the limit of $\delta\theta \to 0$ [21], the gain of the ATAN2 can be calculated by:

$$\begin{aligned} K_D^{A\tan 2} &= \frac{\partial \mu_e}{\partial \delta_\theta}|\delta_\theta = 0 \\ &= \int_{-\pi}^{\pi} \phi P'(\phi)d\phi \end{aligned} \tag{18}$$

where $P'(\phi)$ denotes the first derivative of $P(\phi)$, and it is expressed as:

$$\begin{aligned} P'(\phi) &= \frac{e^{-\frac{SNR}{2}}}{4\pi}\sqrt{SNR}\sin(\phi) \times (\sqrt{2\pi}e^{\frac{1}{2}SNR\cos(\phi)^2}(SNR\cos(\phi)^2) + 1) \times \\ &\left(erf\left[\frac{SNR\cos(\phi)}{\sqrt{2}}\right] + 1\right) + 2\sqrt{SNR}\cos(\phi)) \end{aligned} \tag{19}$$

Similarly, the variance of the ATAN2 can be expressed as [21]:

$$\begin{aligned} Var(n^\theta) &= \int_{-\pi}^{\pi} \arctan(\cos(\phi), \sin(\phi))^2 P(\phi)d\phi \\ &= \int_{-\pi}^{\pi} \phi^2 P(\phi)d\phi \end{aligned} \tag{20}$$

Similar to the FLL, in the case of the PLL, the PDF obeys the Gaussian distribution [21] at extremely high SNR, so the variance can be expressed by:

$$Var(n^\theta) = \frac{2}{SNR}\left(rad^2\right) \tag{21}$$

Meanwhile, at a very low SNR, Equation (20) tends to the uniform distribution within the interval of $[-\pi, \pi]$ [21], which is a theoretical upper bound of Equation (20). For the PLL, the mean square error of three times of the phase measurement error should not exceed one fourth of the phase discrimination pull-in range, in other words, i.e., $3\sigma_{PLL} \leq \frac{\pi}{4}$ [17], where $\sigma_{PLL}^2$ represents $Var(n^\theta)$ in this paper. Similarly, the PLL tracking threshold is taken as the variance under low SNR. After squaring both sides of the inequality $3\sigma_{PLL} \leq \frac{\pi}{4}$, so it holds that:

$$9Var\left(n^\theta\right) \leq \frac{\pi^2}{16} \tag{22}$$

### 2.3.2. Filter Coefficient Determination

It is necessary to determine coefficients $A_0$ and $A_1$ of the second-order filter of the PLL. However, the coefficients of the PLL and FLL are different. In the following, the derivation of coefficients of the PLL is provided.

The transmission function (TF) between carrier phase $\theta$ and carrier phase estimate $\hat{\theta}$ is given by:

$$H_\theta(z) = \frac{K_D F(z) NCO(z)}{1 + K_D F(z) NCO(z)} \tag{23}$$

where $F(z)$ denotes the filter function, and $NCO(z)$ represents the NCO function. The expression of the second-order filter is as follows [22]:

$$F(z) = A_0 + A_1 \frac{z T_L}{z - 1} \tag{24}$$

Thus, $H_\theta(z)$ can be simplified to:

$$H_\theta(z) = \frac{1}{1 + K_D \cdot \left(A_0 + A_1 \cdot \frac{z \cdot T_L}{z-1}\right) \cdot \left(\frac{z \cdot T_L}{z-1}\right)} \tag{25}$$

The pole represents the solution to Equation (25) when the denominator of the TF is zero. Thus, the two poles of Equation (25) are expressed as follows:

$$\begin{aligned} z_a &= \frac{A_0 K_D T_L + 2 + \sqrt{(A_0 K_D T_L)^2 - 4 A_1 K_D T_L^2}}{2(A_1 K_D T^2 + A_0 K_D T_L + 1)} \\ z_b &= \frac{-A_0 K_D T_L - 2 + \sqrt{(A_0 K_D T_L)^2 - 4 A_1 K_D T_L^2}}{2(A_1 K_D T^2 + A_0 K_D T_L + 1)} \end{aligned} \tag{26}$$

The two poles can be represented as $z_a = e^{-\beta(1+\eta)}$ and $z_b = e^{-\beta(1-\eta)}$ [1], therefore, the values of two filter coefficients can be obtained as:

$$\begin{aligned} A_0 &= -\frac{2 e^{-\beta} - e^{-\beta(1-\eta)} - e^{-\beta\eta}}{K_D T_L e^{-\beta}} \\ A_1 &= \frac{e^{-\beta} - e^{-\beta(1-\eta)} - e^{-\beta\eta} + 1}{K_D T_L^2 e^{-\beta}} \end{aligned} \tag{27}$$

The values of $\beta$ and $\eta$ are given in the FLL analysis. The noise bandwidth of the system $H_\theta(s)$ is defined as:

$$B_\omega = \frac{1}{2\pi T_L} \int_{-\pi}^{\pi} \left| H_\theta\left(e^{i\omega}\right) \right|^2 d\omega \tag{28}$$

By applying the Cauchy residue theorem [1] to Equation (28), we get [21]:

$$B_\omega = \frac{2\frac{A_1}{A_0} + 2 A_0 K_D + A_1 K_D T_L}{4 - K_D T_L (2 A_0 + A_1 T_L)} \tag{29}$$

According to Equations (28) and (29), the tracking variance of the loop can be expressed as:

$$\begin{aligned} \sigma_{\delta\theta}^2 &= \frac{Var(n^\theta)}{2\pi} \int_{-\pi}^{\pi} \left| H_n\left(e^{i\omega}\right) \right|^2 d\omega = \frac{Var(n^\theta)}{2\pi K_D^2} \int_{-\pi}^{\pi} \left| H_\theta\left(e^{i\omega}\right) \right|^2 d\omega = \frac{Var(n^\theta)}{K_D^2} B_\omega \\ &= \frac{Var(n^\theta)}{K_D^2} \cdot \frac{2\frac{A_1}{A_0} + 2 A_0 K_D + A_1 K_D T_L}{4 - K_D T_L (2 A_0 + A_1 T_L)} \end{aligned} \tag{30}$$

where $H_n\left(e^{i\omega}\right)$ denotes the noise transfer function.

By simplifying Equation (30), we get:

$$\sigma_{\delta\theta}^2 = \frac{c_1 Var(n^\theta)}{c_2 K_D^2} \tag{31}$$

where $c_1 = 2\frac{A_1}{A_0} + 2A_0K_D + A_1K_DT_L$, and $c_2 = 4 - K_DT_L(2A_0 + A_1T_L)$.

### 2.3.3. PLL Segmentation Results

Similar to the FLL segmentation, the first segmentation point is obtained by substituting Equations (20) and (21) into Equation (30) in turn. For the PLL, the theoretical variance deviates from the variance at the high CNR when CNR is 51.125 dB-Hz under the conditions of an integration time of 2 ms and a bandwidth of 25 Hz. When Equations (20) and (22) are substituted into Equation (30), a new equation will be formed, and then the value of the second segmentation point can be obtained.

The tracking variance can be expressed by segmented functions as follows:

$$\sigma_{\delta\theta}^2 = \begin{cases} \frac{c_1}{c_2 SNR K_D^2} & \text{CNR} > 51.125 \text{ dB-Hz} \\ \frac{c_1 \pi}{12 c_2 K_D^2} & \text{CNR} < 27.412 \text{ dB-Hz} \end{cases} \tag{32}$$

## 3. Segmentation Point-Affecting Factors

Equations (10) and (32) show the difference in the segmentation results between the FLL and PLL, so the factors affecting the segmentation point should be discussed. The segmentation results of the PLL and FLL are verified by simulation.

### 3.1. Simulation Verification of Segmentation Points

In order to verify equations of segmentation results, the FLL and PLL were simulated.

### 3.1.1. FLL Simulation Verification

In order to verify the accuracy of the presented theoretical derivation, the simulation was performed using the receiver that was designed in MATLAB software. In the simulation, the integration time was set to 2 ms, and the loop noise bandwidth was 25 Hz, which is an empirical value commonly used in simulation. In the simulation, the satellite was moving all the time, but the additional dynamic acceleration was not considered since it is commonly used in simulation conditions of an open environment. The signal strength was in the range of 20–55 dB-Hz; this range was selected because it is suitable for GNSS receivers. The PRN12 satellite data were used for verification.

The variance curve of the FLL is presented in Figure 2. It was obtained by substituting Equation (1) into Equation (5), while the variance curve of a high SNR was obtained by substituting Equation (3) into Equation (5). The variance curve of a low SNR was obtained by substituting Equation (4) into Equation (5).

As presented in Figure 2, the simulation results were in good agreement with the theoretical results. In Figure 2, the content of the dashed box is enlarged and presented in the inset given at the lower-left corner to demonstrate that the gap between the two curves begins to widen at 48.496 dB-Hz. The variance curve of high SNR began to diverge from the variance curve at 48.496 dB-Hz, so the signal was regarded as a strong signal when the signal strength was greater than 48.496 dB-Hz, and when the signal strength was smaller than 25.079 dB-Hz, it was regarded as a weak signal because the variance reached the threshold. Generally speaking, the signal with a strength greater than 40 dB-Hz can be regarded as a strong signal, while the signal with a strength smaller than 28 dB-Hz can be regarded as a weak signal [17]. The result presented in Figure 2 is more specific than the empirical values given in reference [17]. Also, the result in Figure 2 represents the segmentation result of particular integration time and bandwidth, which is a special case. The influence of the integration time and bandwidth on the segmentation results will be analyzed in Section 3.2.

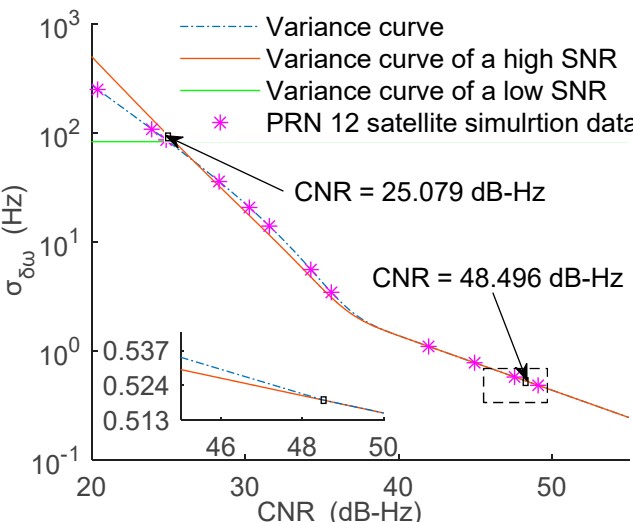

**Figure 2.** Segmentation result of the FLL under the integration time of 2 ms and the filter bandwidth of 25 Hz.

3.1.2. PLL Simulation Verification

The segmentation result of the PLL is shown in Figure 3, the variance curve of high SNR was obtained by substituting Equation (21) into Equation (30), the variance curve of low SNR was obtained by substituting Equation (22) into Equation (30), and the variance curve was obtained by substituting Equation (20) into Equation (30). The simulation data were obtained under the same simulation conditions as that of the FLL.

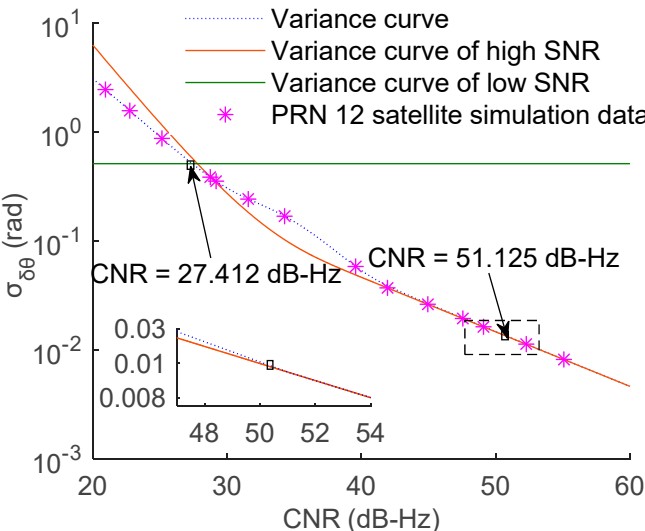

**Figure 3.** Segmentation result of the PLL under the integration time of 2 ms and the filter bandwidth of 25 Hz.

According to the results presented in Figure 3, the simulation and theoretical derivation results of the PLL results were in good agreement. The content of the dashed box is enlarged and displayed in the inset at the lower-left corner to demonstrate that for the PLL, the gap between the two curves begins to widen at 51.125 dB-Hz. However, there were certain differences in the segmentation result between the PLL and FLL, which will be explained in the following.

### 3.1.3. Difference between FLL and PLL Segmentation Points

The segmentation points represent the respective strong and weak signal ranges of the two loops, showing the difference in performances between the FLL and PLL. The segmentation points of the two loops were compared and analyzed. The values of the segmentation points for an integration time of 2 ms and a bandwidth of 25 Hz are given in Table 1 and displayed in Figure 4.

**Table 1.** Comparison of the segmentation results of the FLL and PLL with the four-quadrant arctangent discriminator.

|  | Strong Signal (dB-Hz) | Weak Signal (dB-Hz) |
|---|---|---|
| FLL | CNR $\geq$ 48.496 | CNR $\leq$ 25.079 |
| PLL | CNR $\geq$ 51.125 | CNR $\leq$ 27.412 |

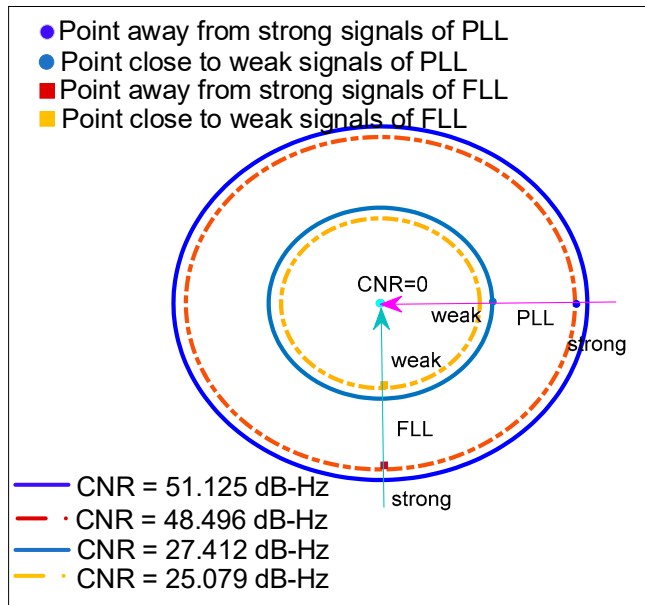

**Figure 4.** Differences in segmentation results between FLL and PLL.

As shown in Table 1 and Figure 4, in the process of signal strength changing from strong to weak (refer to the direction of the arrow in Figure 4, the point that was away from strong signals of the PLL was earlier than that of the FLL, and the point close to weak signals of the PLL was also earlier than that of the FLL. Thus, the PLL was more sensitive to the signal intensity, and at a low CNR, it failed earlier than the FLL. The FLL operated well even at a low CNR [23]. The PLL had the out-of-lock condition at about 27 dB-Hz, while the FLL was about 25 dB-Hz. This was the reason why the FLL could track weaker signals than the PLL.

### 3.2. Common Factors Affecting Segmentation Results

According to Equations (5) and (30), the integration time and bandwidth are common factors affecting segmentation results. In the following, their effects on the segmentation results will be analyzed in detail.

### 3.2.1. Effect of Integration Time

In this section, the segmentation results of the FLL and PLL are compared at different integration times. The integration time was set to 2, 4, and 10 ms in turn; these values were selected because they are commonly used in simulations. The FLL segmentation results at different integration times are presented in Figure 5.

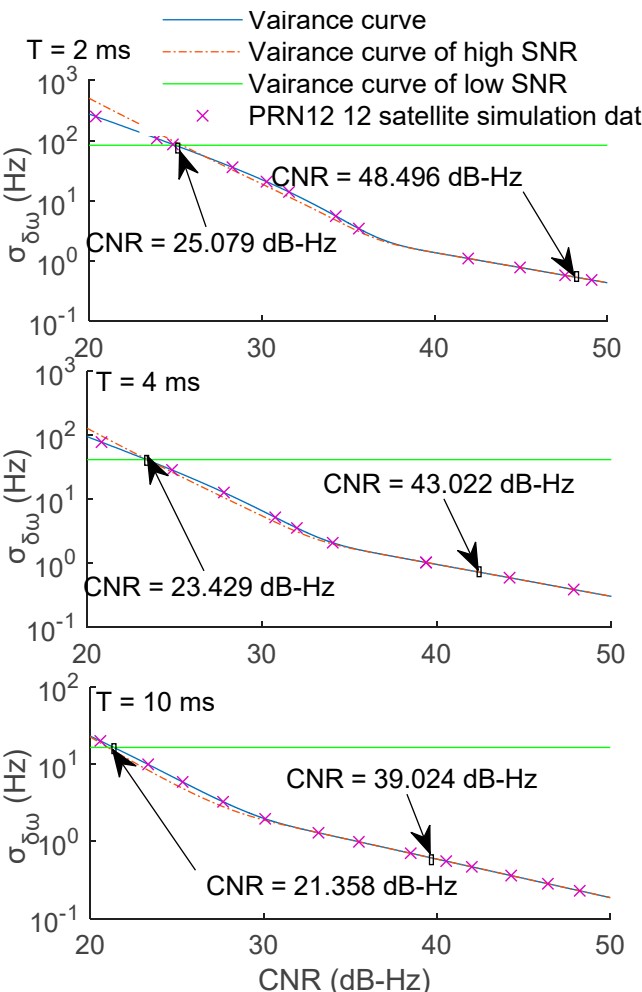

**Figure 5.** Segmentation results of the FLL at different integration times.

As shown in Figure 5, the variance decreased with the integration time. In the signal changing process from strong to weak, the integration time increased, and the segmentation points appeared more and more "late", so even a weak signal could be tracked. Thus, increasing the integration time can be beneficial to enhancing the loop performance at a low CNR. However, the integration time cannot be increased infinitely because too long integration time can lead to the reversal of data bits. Therefore, in practice, the integration time generally should not exceed 20 ms.

As for the PLL, with the increase in the integration time, the changing trend of segmentation points was the same as that of the FLL, which is why only the change in segmentation points of the FLL is displayed in Figure 5. The results of the PLL and FLL are summarized in Table 2 and displayed in Figure 6.

**Table 2.** Influence of the integration time on the segmentation results.

|  | FLL | | PLL | |
|---|---|---|---|---|
|  | Weak Signal (dB-Hz) | Strong Signal (dB-Hz) | Weak Signal (dB-Hz) | Strong Signal (dB-Hz) |
| $T = 2$ ms | CNR $\leq$ 25.079 | CNR $\geq$ 48.496 | CNR $\leq$ 27.412 | CNR $\geq$ 51.125 |
| $T = 4$ ms | CNR $\leq$ 23.429 | CNR $\geq$ 43.022 | CNR $\leq$ 26.026 | CNR $\geq$ 46.887 |
| $T = 10$ ms | CNR $\leq$ 21.358 | CNR $\geq$ 39.024 | CNR $\leq$ 24.716 | CNR $\geq$ 42.853 |

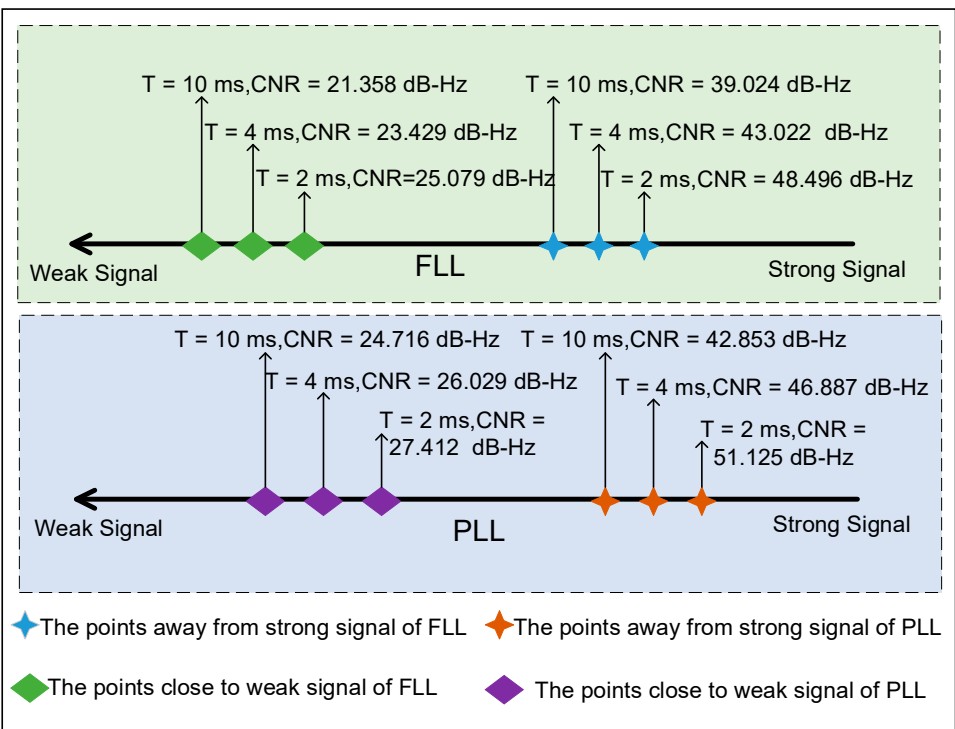

**Figure 6.** Difference in segmentation results between the FLL and PLL at different integration times.

Based on the results in Table 2 and Figure 6, the value of all segmentation points changed with the integration time, and the segmentation results of the PLL and FLL were still different. The reasons for this difference will be analyzed in detail following the following.

### 3.2.2. Effect of Bandwidth

The bandwidth ($B_\omega$) is another important affecting parameter, and its value should be chosen so that to achieve a balance between low noise and high dynamic performance [17]. The bandwidth influence on the segmentation results was analyzed by setting a fixed integration time of 2 ms, and the bandwidth to 2, 10, and 25 Hz, in turn. The segmentation results of the FLL under different bandwidth conditions are presented in Figure 7.

As presented in Figure 7, the increase in the bandwidth resulted in the increase in the loop variance and "earlier" appearance of segmentation points. Therefore, when the dynamic performance of the loop was not taken into account, a small bandwidth value was beneficial to the loop performance. Similarly, the trend was the same as that of the PLL, which is why only the FLL result is shown in Figure 7. The segmentation results of the PLL and FLL under different bandwidths are summarized in Table 3 and displayed in Figure 8.

**Table 3.** Influence of the filter bandwidth on the segmentation results.

| | FLL | | PLL | |
|---|---|---|---|---|
| | **Weak Signal (dB-Hz)** | **Strong Signal (dB-Hz)** | **Weak Signal (dB-Hz)** | **Strong Signal (dB-Hz)** |
| $B\omega$ = 2 Hz | CNR $\leq$ 19.567 | CNR $\geq$ 43.343 | CNR $\leq$ 22.330 | CNR $\geq$ 48.722 |
| $B\omega$ = 10 Hz | CNR $\leq$ 23.128 | CNR $\geq$ 46.040 | CNR $\leq$ 25.453 | CNR $\geq$ 49.027 |
| $B\omega$ = 25 Hz | CNR $\leq$ 25.079 | CNR $\geq$ 48.496 | CNR $\leq$ 27.412 | CNR $\geq$ 51.125 |

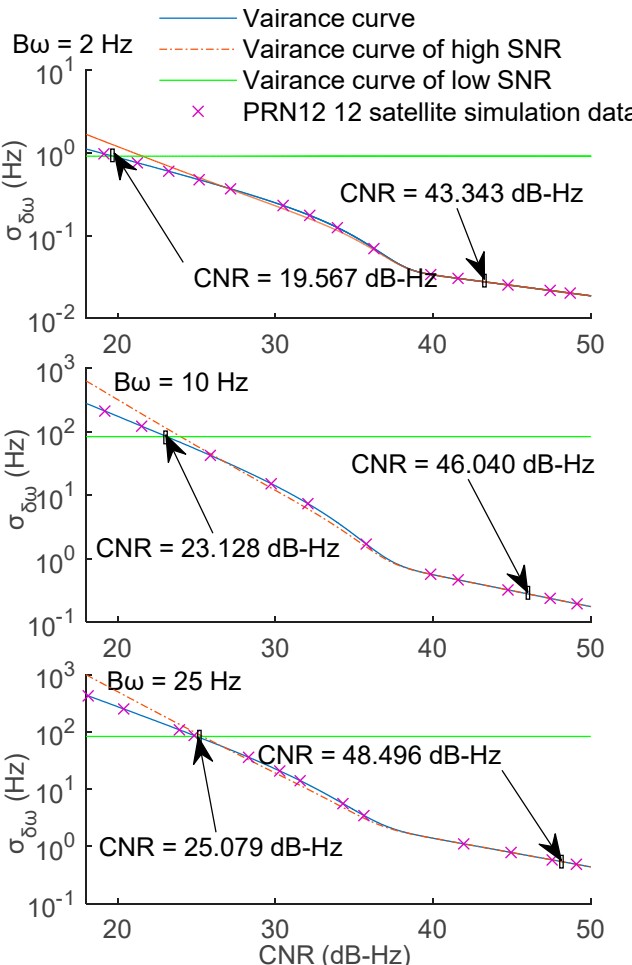

**Figure 7.** Segmentation results of the FLL at different bandwidths.

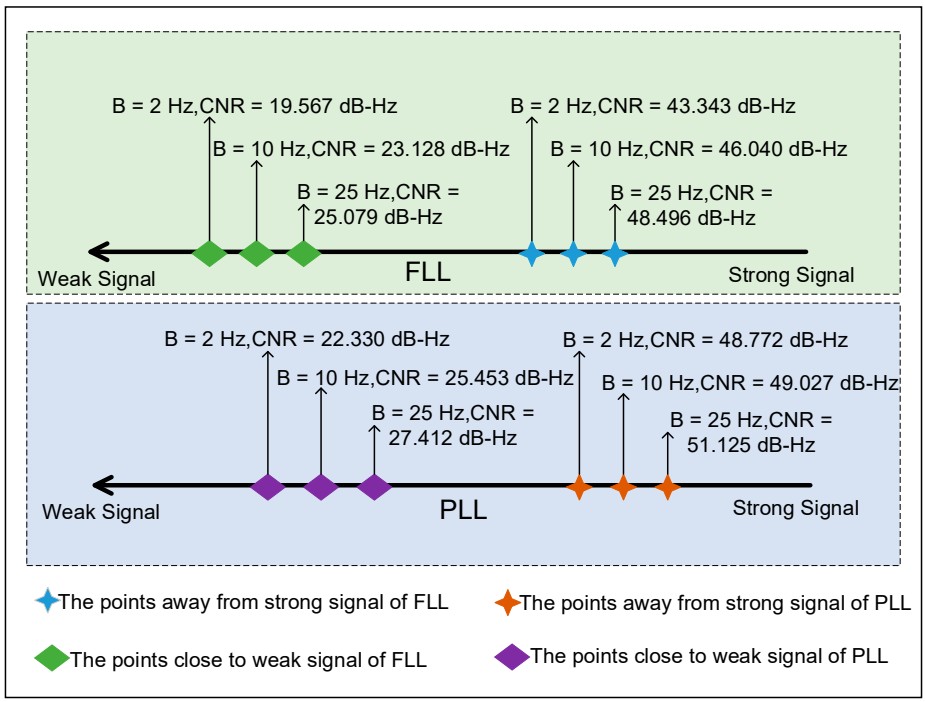

**Figure 8.** Difference in segmentation results between the FLL and PLL at different bandwidth.

As presented in Table 3 and Figure 8, with the increase in the bandwidth, the changing trends of the FLL and PLL segmentation points were the same, but at the same bandwidth, there were certain differences in the segmentation results of the two loops. The factors causing these differences are analyzed in the following.

### 3.3. Factors Causing Differences in Segmentation Results

According to the segmentation results obtained by Equations (10) and (32), even when discriminators' integration times and bandwidths were the same for the two loops, the segmentation results of the FLL and PLL were still different. According to Equations (5) and (30), the main difference between the FLL and PLL lies in coefficients $A_0$ and $A_1$ and gain $K_D$. The specific analysis is provided in the following.

#### 3.3.1. Effect of Discriminator Gain

The differences between the phase discrimination gain and frequency discrimination gain are shown in Figure 9.

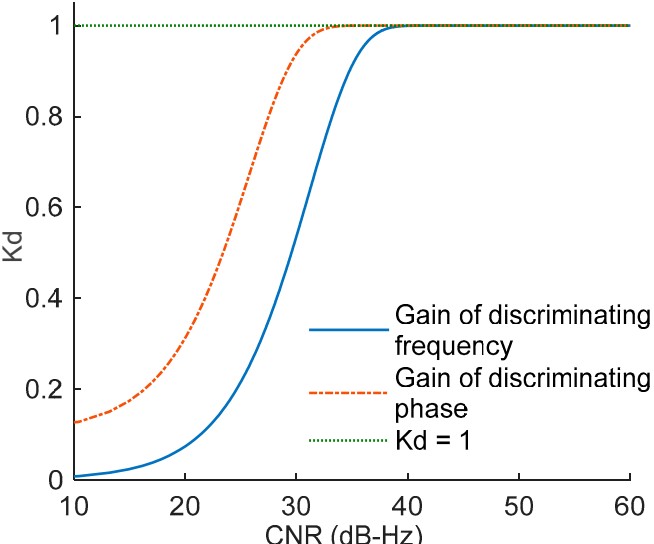

**Figure 9.** The difference in gain between the discriminating phase and discriminating frequency of four-quadrant arctangent discriminator.

As shown in Figure 9, the gain of both PLL and FLL decreased with the decrease in the CNR, but the FLL gain decreased earlier than that of the PLL. Therefore, in the signal weakening process, the FLL was more responsive than the PLL. Under the same signal conditions, the gain of the discriminating frequency was less than that of the discriminating phase, so the PLL achieved higher gain and accuracy than the FLL. Since the gain has a certain compensation effect on the subsequent filter coefficient, a different gain leads to the difference in the filter coefficient.

#### 3.3.2. Influence of Filter Coefficients

The filter coefficients of the PLL and FLL were compared and analyzed. The different changing trends in $A_0$ between the PLL and FLL is presented in Figure 10. In the analysis, the integration time was set to 2 ms and the bandwidth was set to 25 Hz.

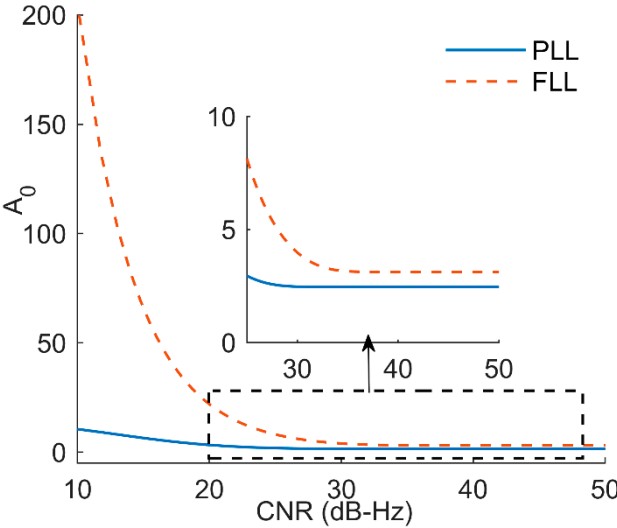

**Figure 10.** The changing trends of coefficient $A_0$ of the PLL and FLL.

The coefficient $A_0$ represents the proportion parameter of a filter, so the main aim is to speed up system response. The larger the values of coefficient $A_0$ is, the faster the system response will be. As shown in Figure 10, the coefficient $A_0$ of the FLL was larger than that of the PLL, which showed that the FLL could react faster and achieved better dynamic performance than the PLL [24]. The coefficient $A_1$ of the PLL and FLL is shown in Figure 11.

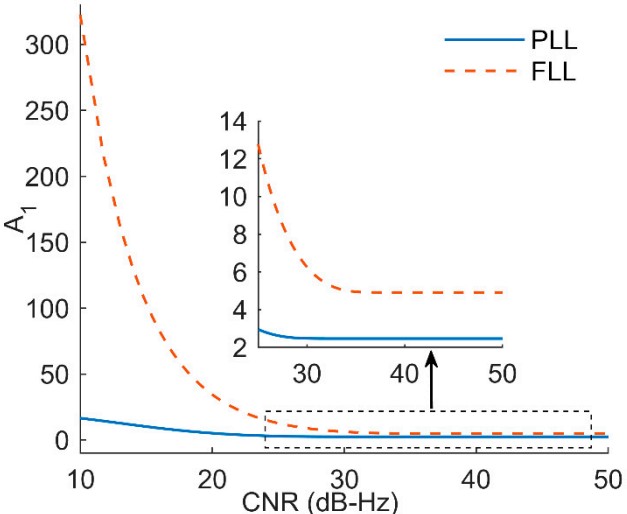

**Figure 11.** The changing trends of coefficient $A_1$ of the PLL and FLL.

The coefficient $A_1$ represents the integral parameter of a filter, which can be used to eliminate the steady-state error of the system. The coefficient $A_1$ of the FLL was larger than that of the PLL, which showed that the FLL had a larger tracking error than the PLL; thus, a larger value of coefficient $A_1$ was needed to eliminate the error. Therefore, the difference in performances of the two loops is closely related to the two coefficients.

Consequently, coefficients $A_0$ and $A_1$ have a great influence on the poles. The pole-zero maps of the FLL and PLL are presented in Figures 12 and 13, respectively.

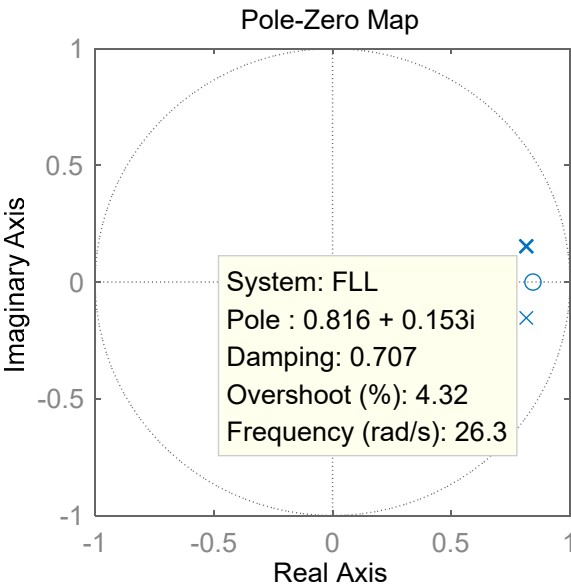

**Figure 12.** Pole-zero map of the second-order FLL.

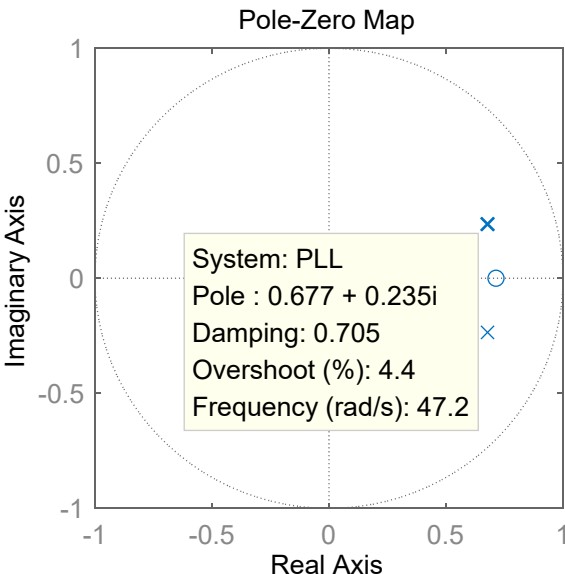

**Figure 13.** Pole-zero map of the second-order PLL.

As shown in Figures 12 and 13, the poles of both PLL and FLL were located within the unit circle, indicating that the second-order PLL and FLL were stable. However, the pole of the PLL was closer to the imaginary axis, indicating that the oscillation time was longer and the response was slower than those of the FLL.

## 4. Discussion and Validation with Real Data

Based on the above-presented analysis results, the filter coefficients still have many aspects worthy of further studying, so they are discussed in the following. An increase in the integration time and a reduction in the bandwidth can reduce the tracking error, but they are limited. So, it is necessary to discuss the relationship between the integration time and bandwidth. The normalized bandwidth and phase margin were analyzed and coefficients $A_0$ and $A_1$ were verified by the real data.

### 4.1. Discussion on Filter Coefficients

Two key points in this paper are the tracking loop segmentation and filter coefficients. In the previous analysis, it is shown that the segmentation results are affected by the filter bandwidth and integration time. therefore, the purpose of this discussion is to examine whether the filter coefficients are also affected by these two factors, and on this basis, the phase margin (PM) and subsequent stability analysis results are discussed.

The normalized bandwidth ($B_\omega T_L$) denotes an important parameter in determining the filter coefficients. Therefore, the relationship between the filter coefficients and error was analyzed at different $B_\omega T_L$ values, and obtained results are presented in Figures 14 and 15, where the CNR is 40 dB-Hz.

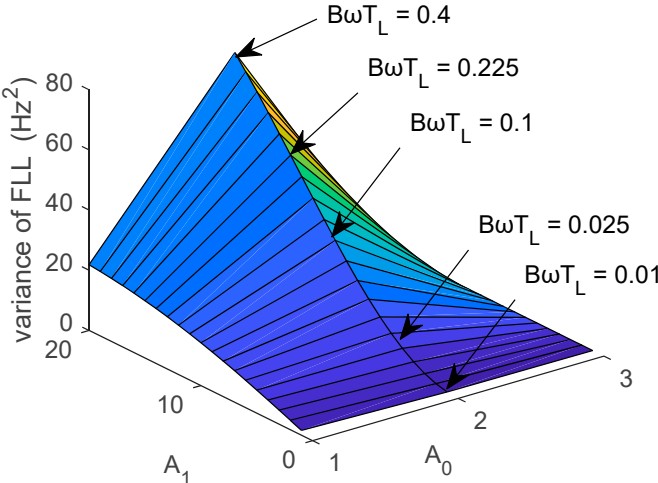

**Figure 14.** Change in the variance of the FLL at different normalized bandwidths.

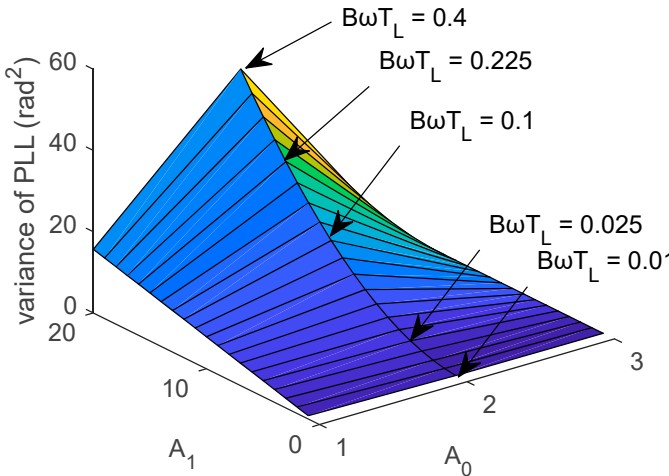

**Figure 15.** Change in the variance of the PLL at different normalized bandwidths.

As shown in Figures 14 and 15, when $B_\omega T_L$ decreased, the variance also decreased. Also, the value of $B_\omega T_L$ had a certain impact on the filter coefficients $A_0$ and $A_1$.

In order to analyze the relationship between $B_\omega$ and $T_L$ in detail, the concept of PM was used. The PM was defined as 180 degrees plus the phase of the open-loop at the frequency of the loop bandwidth when the loop gain was unity, which was suitable for both the FLL and the PLL. The PMs of the first-, second- and third-order PLLs have been analyzed in detail in [25]. Therefore, in this work, the PM of the second-order FLL was calculated according to the results of [25]. The difference between the PLL and FLL in

calculating the PM is the loop coefficient. According to the TF in the *z*-domain, the loop coefficients can be expressed as [25,26]:

$$G_1 = 2\xi\omega_n - \frac{\omega_n^2 T}{2}, G_2 = \omega_n^2 T \quad \text{(PLL)} \tag{33}$$

$$G_1 = 2\xi\omega_n T, G_2 = \omega_n^2 T^2 \quad \text{(FLL)} \tag{34}$$

where $\xi$ denotes the damping coefficient, and $\omega_n$ denotes the characteristic frequency and for the second-order loop, it holds that $\omega_n = \frac{8B_\omega\xi}{1+4\xi^2}$ [27]; $T$ denotes the integration time. Thus, the PM represents a function of $B_\omega$ and $T$. There is difference between the loop coefficients in Equations (33) and (34) and the loop coefficients in Equations (8), (9) and (27). The loop coefficients in Equations (33) and (34) were transformed from the *s*-domain into the *z*-domain by the bilinear transform, while the loop coefficients in Equations (8), (9) and (27) are obtained from the zero and pole of the TF and in the form of a second-order (proportional and integral (PI)) style controller.

The PM is an important measure of the feedback system stability. The larger the PM is, the more stable the loop and the more allowable the deviation will be. The relationships between the PM and the bandwidth $B_\omega$ and integration time $T$ of the FLL and PLL are presented in Figures 16 and 17, respectively, where $\xi = 0.707$.

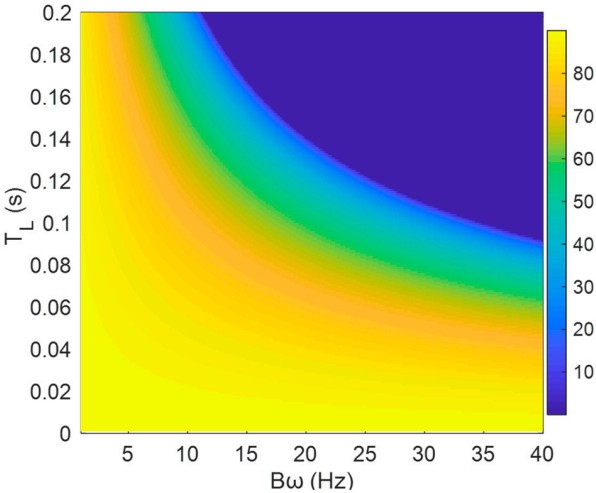

**Figure 16.** The PM of the second-order FLL.

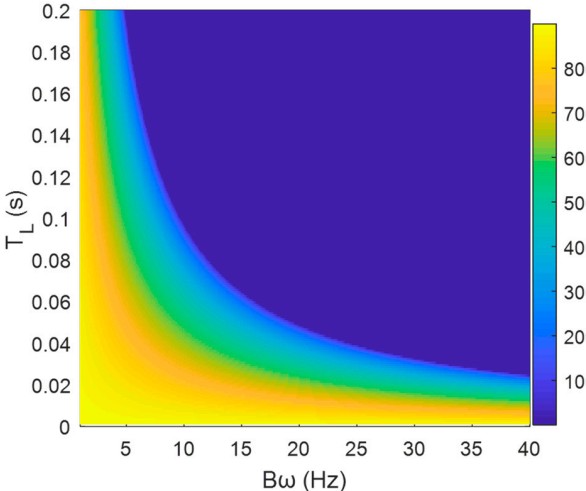

**Figure 17.** The PM of the second-order PLL.

According to the results presented in Figures 16 and 17, with the increase in $B_\omega T_L$, the PM decreased, and the stability degraded. At the same $B_\omega T_L$ value, the PM of the FLL was greater than that of the PLL, which reflected the performance difference between the two loops. The FLL was more stable than the PLL and also could operate better at a higher noise level.

At a narrow bandwidth, an increase in the integration time did not have a significant impact on loop stability. However, when the bandwidth was too wide, by increasing the integration time, the PM reduced, and the two loops' stabilities decreased. Thus, the bandwidth affected the loops' performance significantly. In a certain range of the integration time, such as 2–10 ms, the increase in the bandwidth did not have a significant effect on system stability. At this time, an appropriate value of the integration time could ensure loop stability, but when the integration time exceeded a certain value, the increase in the bandwidth caused a decline in loop stability. Accordingly, it can be concluded that there is a dynamically balanced relationship between $B_\omega$ and $T_L$, so there should be a limit on the normalized bandwidth $B_\omega T_L$ to ensure good loop stability. Generally, the PM of the loop should be greater than thirty degrees. For instance, at $\xi = 0.707$, the limit of $B_\omega T_L$ of the second-order PLL is about 0.46 [25].

### 4.2. Validation with Real Data

In order to validate derived equations, the actual data obtained from the GPS receiver were used. The GPS receiver was set in an open environment near the East Lake of Hohai University, as shown in Figure 18. The sampling frequency was 16.396 MHz. Three observations were made, and each observation interval lasted for about five minutes. The actual raw data were post-processed using the MATLAB software.

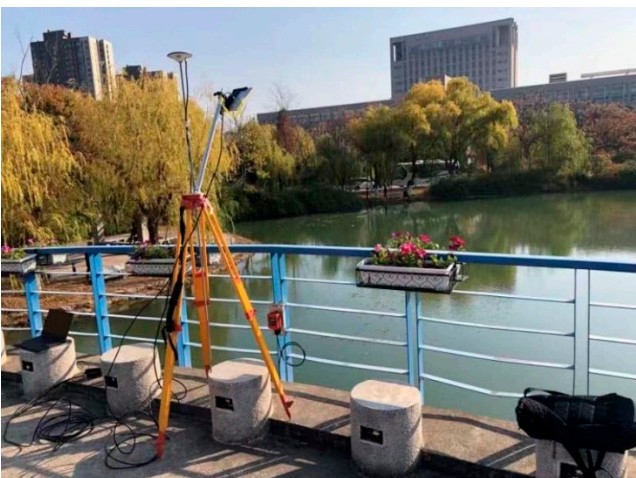

**Figure 18.** The GPS receiver used for collecting real data at the Hohai University.

As shown in the top, middle and bottom of Figure 19, in the first, second, and third observations, five, six, and seven signal channels were acquired, respectively.

Since the FLL has been analyzed in detail by Curran et al. [1] and due to the limited length of the article, only the PLL results are used for validation. The comparisons between the real and theoretical data of coefficients $A_0$ and $A_1$ are presented in Figures 20 and 21, respectively.

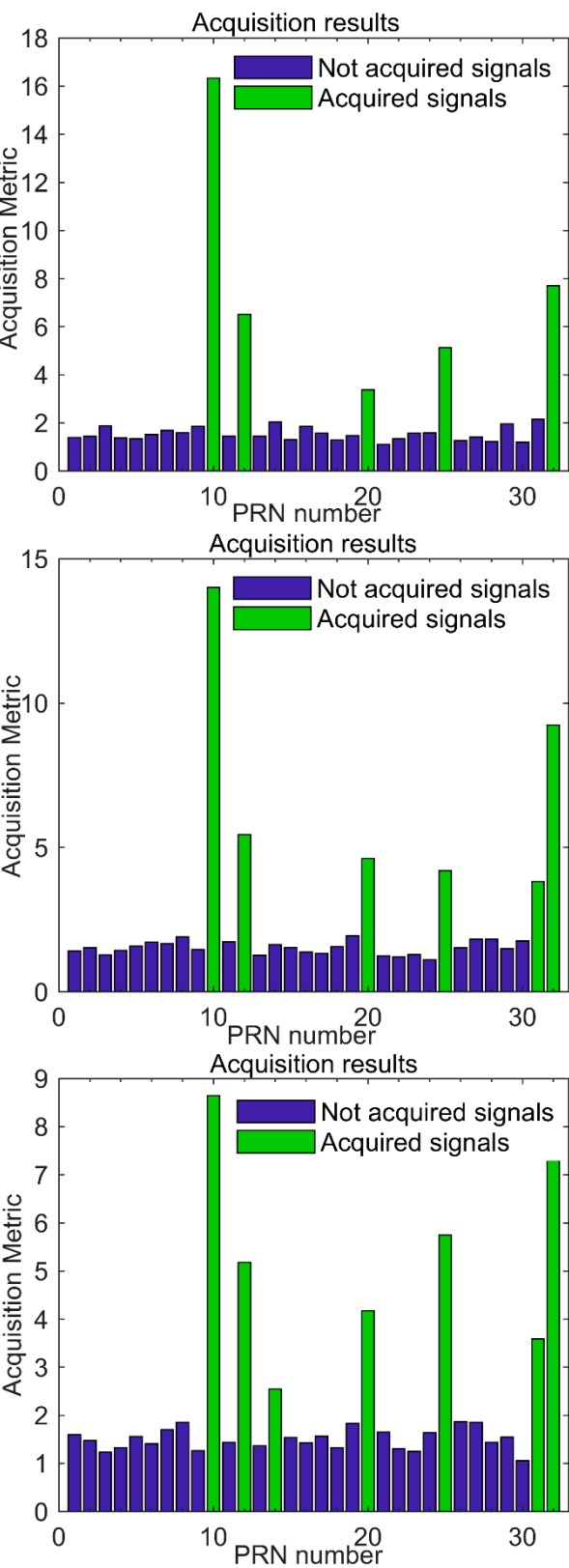

**Figure 19.** Acquisition results of three actual observations.

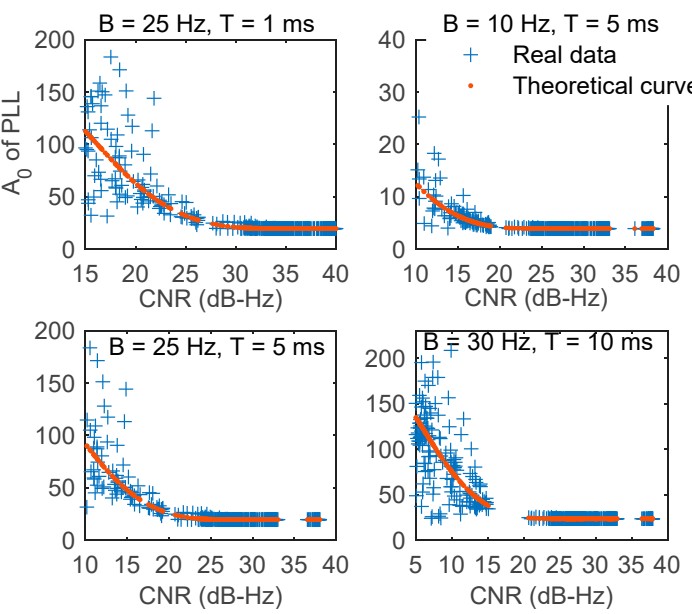

**Figure 20.** Real and theoretical data of $A_0$.

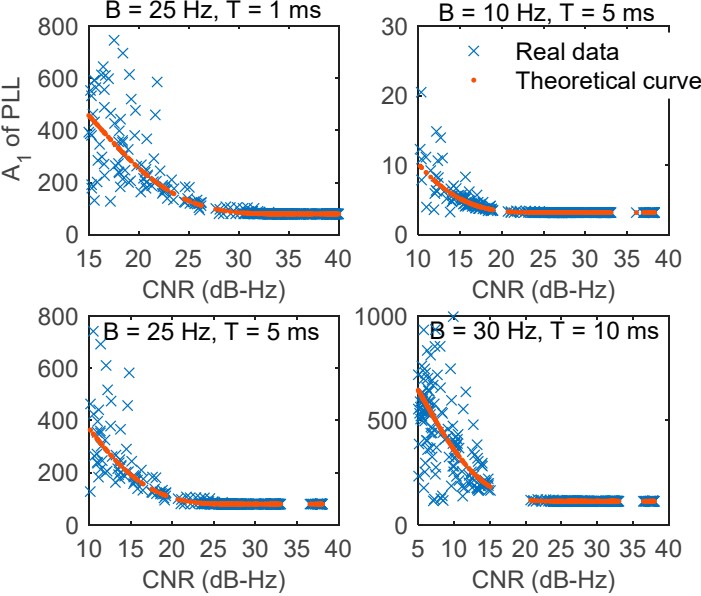

**Figure 21.** Real and theoretical data of $A_1$.

As shown in Figures 20 and 21, the real data of $A_0$ and $A_1$ deviated slightly from the theoretical values under different normalized bandwidth values. The numerical values of the deviation degree are given in Table 4, where $\Delta A_0$ represents the difference between the real and theoretical values of $A_0$, and $\Delta A_1$ represents the difference between the real and theoretical values of $A_1$.

**Table 4.** Standard deviation of the error between the real and theoretical data.

|  | $B_\omega = 25$ Hz, $T = 1$ ms | $B_\omega = 10$ Hz, $T = 5$ ms | $B_\omega = 25$ Hz, $T = 5$ ms | $B_\omega = 30$ Hz, $T = 10$ ms | $B_\omega = 50$ Hz, $T = 15$ ms |
|---|---|---|---|---|---|
| $\Delta A_0$ | 4.24 | 1.78 | 4.25 | 4.48 | 6.23 |
| $\Delta A_1$ | 8.56 | 1.60 | 8.54 | 10.00 | 17.17 |

As presented in Figures 20 and 21 and Table 4, with the increase in the CNR, the coincidence effect improved, and the deviation gradually reduced, which was because with the increase in the CNR, the discriminator gain increased slowly, and the value gradually reached a value of one. When the CNR deviated slightly, the changes in $A_0$ and $A_1$ were small. When the bandwidth was in an appropriate range, increasing the integration time and reducing the bandwidth were beneficial to the loop stability. This effected can be observed in Table 4 for the following three set of values: $B_\omega$ = 25 Hz, $T$ = 1 ms, $B_\omega$ = 10 Hz, $T$ = 5 ms, and $B_\omega$ = 25 Hz, $T$ = 5 ms. However, when the bandwidth was too large, the normalized bandwidth widened, and the loop error gradually increased, which can be observed in Table 4 for $B_\omega$ = 30 Hz, $T$ = 10 ms and $B_\omega$ = 50 Hz, $T$ = 15 ms. Therefore, the integration time and bandwidth should be balanced dynamically, and the normalized bandwidth must be limited to ensure the stability of the loop.

## 5. Conclusions

Analyzing the probability distribution of the tracking variance and strict derivation based on the tracking loop theory, an accurate quantitative segmentation method is developed. In practice, the segmentation results obtained by the proposed method can be used for a priori segmentation, which is convenient to achieve robust segmentation directly in the signal domain, rather than using the traditional passive fuzzy adjustment loop segmentation strategy through noise variance feedback.

The contributions of this work can be summarized as follows:

(1) The concept of the forward loop segmentation is introduced, and the segmentation results of the PLL and FLL are analyzed based on the characteristics of variance under different SNR. The difference in the segmentation results indicates the performance difference between the PLL and FLL, which demonstrates that the FLL can track weaker signals than the PLL. The segmentation results of both the theoretical derivation and simulation show that FLL can track about 2.5 dB-Hz more weaker signals than the PLL under the integration time of 2 ms and the filter bandwidth of 25 Hz;

(2) The main reasons for the performance difference between the two loops are the discriminator gain and filter coefficients. In the discriminator stage, the FLL has only about a 0.2 dB-Hz advantage over the PLL, but this advantage increases to 2.5 dB-Hz when the discrimination gain is combined with the subsequent filtering. Therefore, the difference in performance between the two loops is caused by the combination of discrimination gain and filtering;

(3) The proportion coefficient ($A_0$) of the FLL is larger than that of the PLL, so the FLL has better robustness and dynamic performance than the PLL. The integration coefficient ($A_1$) of the FLL is also larger than that of the PLL, so the FLL has a larger tracking error than the PLL. The difference is also reflected in the phase margin of the PLL and FLL. Moreover, reducing the normalized bandwidth can reduce the tracking variance of both the PLL and the FLL, but a dynamic balance between the integration time and bandwidth is necessary to achieve the best possible performance. The proposed method directly evaluates the tracking performance from the design factors of a tracking loop.

In future research, a robust loop can be designed by selecting weights according to the loop segmentation results. Moreover, the analysis of discrimination gain and filter coefficient can be used to design improved discrimination and filtering methods.

**Author Contributions:** Q.W. conceived the segmentation idea; M.H. (Mengyue Han) and Q.W. made a theoretical derivation and verified it by simulation; Y.W., X.H., and M.H. (Min He) put forward feasible suggestions on the idea of the article and also suggested changes to the first manuscript and modified the draft. All authors have read and agreed to the published version of the manuscript.

**Funding:** This research was funded by the National Natural Science Foundation of China under Grant Nos. 41474027 and 41974001, and the Key Program of the National Natural Science Foundation of China under Grant Nos. 41830110.

**Data Availability Statement:** The data presented in this study are openly available in FigShare at https://doi.org/10.6084/m9.figshare.13487265.v1.

**Acknowledgments:** We would like to thank the anonymous reviewers and members of the editorial team for their comments and contributions.

**Conflicts of Interest:** The authors declare no conflict of interest.

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
