# Peer review of "Comparison of the Segmentation Results of Two Carrier Tracking Loop Types and Analysis of Theoretical Influencing Factors"

_remotesensing, doi:10.3390/rs13112035_

Round 1

Reviewer 1 Report

A quantitative segmentation method is proposed based on the probability distribution of tracking variance and strict derivation according to tracking loop theory. It provides the possibility of direct segmentation in the signal domain. The work presented in the manuscript would be of interest to the study of tracking loop design.

The authors are suggested to give more detailed captions to the figures in the manuscript.

Author Response

Many thanks for your advice. We added more detailed caption of Figure 1, Figure 2, Figure 3, Figure 4, Figure 6, Figure 8, Figure 9, Figure 10, Figure 11, Figure 12, Figure 13, Figure 14, Figure 15, Figure 16, Figure 17, Figure 18, and Figure 19.

Reviewer 2 Report

Dear Editor,

my comments as below:

  1. all acronyms must explained in the main body of text and abstract. Please correct it.
  2. Abstact must included the major findings from research test, e.g. the obatined results etc.
  3. Introduction, please add more references and position from bibliography.
  4. all symbols from algorithm must be explained in the text. Please correct it.
  5. Figure 20 is illegible, please correct it.
  6. Figure 21 is illegible, please correct it.
  7. References, pleas add more publications from books and articles.

Author Response

Thank you very much for your comments. According to your comments, we have actively corrected the manuscript.

Point 1: all acronyms must explained in the main body of text and abstract. Please correct it.

Response 1: Thanks for your advice. We checked the acronyms of the full manuscript carefully. Then we added full names of PLL and FLL in abstract and full name of GNSS and CNR in introduction when they first appear.

Point 2: Abstact must included the major findings from research test, e.g. the obatined results etc.

Response 2: Thanks for your advice. Based on your comments, we have added some quantitative results description in the abstract. The filtering coefficients lead to the difference of 2.5dB-Hz between FLL and PLL. What’s more, through the real data, The minimum mean error rate of the deviation between the real data and the theoretical data is only 1.8%. These two points have been added to the abstract.

Point 3: Introduction, please add more references and position from bibliography.

Response 3: Thanks for your advice. We carefully read the relevant books and articles, and added three references in the introduction and summarized the main contents.

And we add some summary descriptions of the current research, mainly including the research content and main purpose. The summary descriptions are in lines 40-41, 61-62 and 80-82, respectively.

Point 4: all symbols from algorithm must be explained in the text. Please correct it.

Response 4: Thanks for your advice. We carefully checked the symbols in the whole manuscript and added some new explanations.

In Equation (1), The fai_2 defined as the phase difference caused by the thermal noise at time m, and fai_1 defined as the phase difference caused by the thermal noise at time (m-1).

In Equation(16), I and Q represent the in-phase and the quadrature components of the input signal after correlation demodulation.

Point 5: Figure 20 is illegible, please correct it.

Response 5: Thanks for your advice. We are sorry for the illegible figure, so we adjusted Figure 20 to show the result more clearly.

Point 6: Figure 21 is illegible, please correct it.

Response 6: Thanks for your advice. We adjusted Figure 21 to show the result more clearly.

Point 7: References, pleas add more publications from books and articles.

Response 7: Thanks for your advice. We added three references numbered 4,9,13.

Reviewer 3 Report

Overall, the manuscript is well presented. The organization is clear and the theoretical results are supported by both simulation and real data experiments. However, some equations are not clear. Here are some detailed comments:

  1. Line 108, what's "input-out characteristic"? Confusing term.
  2. Make a distinction when multiplying SNR *T in Eq. (3)
  3. The derivation of Eq. (4) is unclear
  4. How is Eq. (16) is substituted into Eq. (15)? What are I and Q terms?
  5. I’m not sure where Eq. (22) comes from.
  6. Figure 6 is crowded with legends. Trying to have a separate legend on the side. This also applies to other similar figures. 
  7. Table 4 is the deviation, should a different letter used for this such as delta A_0?

Author Response

Thank you very much for your comments. According to your comments, we have actively corrected the manuscript.

In order to see the symbol clearly, the specific modification is in the word file.

Thanks again.
